# Light-Activated Hydroxyapatite Photocatalysts: New Environmentally-Friendly Materials to Mitigate Pollutants

Rafael Lisandro P. Rocha [1,2,3], Luzia Maria C. Honorio [1], Roosevelt Delano de S. Bezerra [2], Pollyana Trigueiro [4], Thiago Marinho Duarte [5], Maria Gardennia Fonseca [6], Edson C. Silva-Filho [1] and Josy A. Osajima [1,*]

1. Interdisciplinary Laboratory for Advanced Materials (LIMAV), UFPI, Teresina 64049-550, PI, Brazil; rafael@ifpi.edu.br (R.L.P.R.); luzia_quimica@yahoo.com.br (L.M.C.H.); edsonfilho@ufpi.edu.br (E.C.S.-F.)
2. Federal Institute of Education, Science and Technology of Piauí, IFPI, Teresina 64000-040, PI, Brazil; rooseveltdsb@ifpi.edu.br
3. Technical-Scientific Police Department, Institute of Criminalistics "Vital Araújo", Teresina 64020-750, PI, Brazil
4. Materials Science and Engineering Postgraduate Program-PPGCM/CCSST, UFMA, Imperatriz 65900-410, MA, Brazil; pollyanatrigueiro@gmail.com
5. Central Analítica, Laboratorio Multi-Usuário, Instituto Nacional Do Semiárido, Campina Grande 58434-700, PB, Brazil; tmd@academico.ufpb.br
6. Fuel and Materials Laboratory (NEP-LACOM), UFPB, João Pessoa 58051-085, PB, Brazil; mgardennia@quimica.ufpb.br
* Correspondence: josyosajima@ufpi.edu.br

**Abstract:** This review focuses on a reasoned search for articles to treat contaminated water using hydroxyapatite (HAp)-based compounds. In addition, the fundamentals of heterogeneous photocatalysis were considered, combined with parameters that affect the pollutants' degradation using hydroxyapatite-based photocatalyst design and strategies of this photocatalyst, and the challenges of and perspectives on the development of these materials. Many critical applications have been analyzed to degrade dyes, drugs, and pesticides using HAp-based photocatalysts. This systematic review highlights the recent state-of-the-art advances that enable new paths and good-quality preparations of HAp-derived photocatalysts for photocatalysis.

**Keywords:** hydroxyapatite; modification strategies; doping; photocatalytic degradation

## 1. Introduction

Worldwide, the contamination of water resources demands management of water consumption and quality [1,2]. Serious environmental consequences have occurred from the intensive use of water resources, population growth, and lack of urban and industrial planning [3–5]. As a result, different classes of molecules used in industrial processes and new compounds that could be contaminants are produced and released into the environment annually [6]. These so-called emerging contaminants include drugs, pesticides, hormones, endocrine disruptors, dyes, and solvents [7,8].

The majority of these compounds are toxic and are not easily destroyed by nature due to their high stability with respect to light and temperature [1], requiring the use of remedial technologies that can mineralize these compounds [9]. To protect the aqueous environment of toxic pollutants, destructive and nondestructive methods include biological methods, chemical precipitation, membrane filtration, electrochemical techniques, reverse osmosis, catalysis, coagulation, electrodialysis, adsorption, and advanced oxidation processes [10–12].

Advanced oxidation processes (AOPs) have been shown to be highly promising for the degradation of organic contaminants, including *Fenton* oxidation, $H_2O_2/UV$, $O_3/UV$, $O_3/H_2O_2/UV$, and photocatalysis [13–15]. AOPs are defined by water treatment processes carried out at room temperature and normal pressure. They are based on the in situ

production of highly reactive hydroxyl radicals ($^{\bullet}$OH) that react non-selectively with most organic compounds and can degrade even highly persistent compounds [16,17].

Among AOPs, a heterogeneous photocatalysis is an efficient tool to purify organic and atmospheric contaminants and convert them into $CO_2$ and $H_2O$ without generating innocuous intermediates under suitable conditions [18,19]. They are based on the use of semiconductor materials that absorb energy from light to generate active species, such as electrons and holes. These are responsible for the formation of active sites on the semiconductor surface that promote the formation of highly oxidized and free radicals capable of degrading various contaminants [20,21]. One of the main advantages of photocatalysis is its high efficiency in modified physicochemical processes, which is recognized as an emerging technology for water treatment [20].

Due to the growing interest in new technologies, a bibliographic search was carried out to find articles in the Scopus database (Title-Abs-Keywords-Advanced) from 2011 to 2021. The keywords—advanced oxidative process, photocatalysis, photocatalyst—found 2534; 62,022; and 55,276 references, respectively, as shown in Figure 1. Data growth was expected because of the growing need to correct environmental problems.

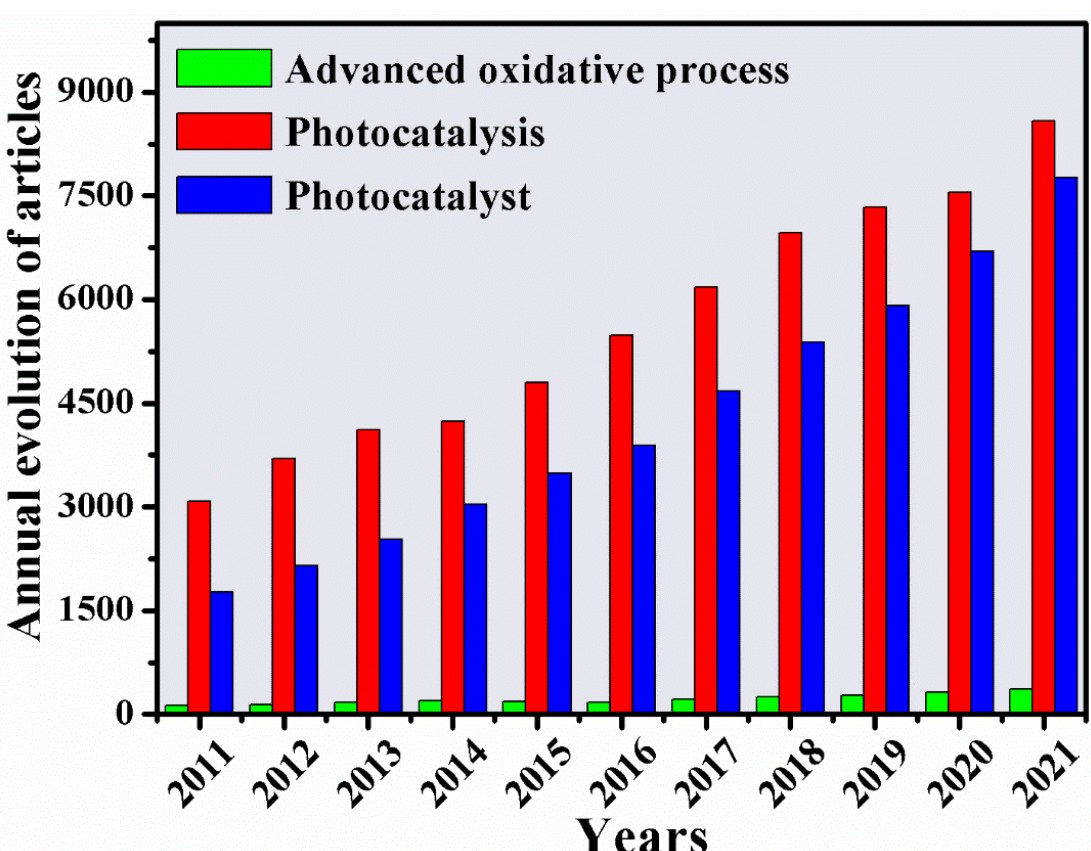

**Figure 1.** Number of annual publications containing the keywords—advanced oxidative process, photocatalysis, and photocatalyst—using the Scopus database.

To mitigate the problems mentioned above, new photocatalysts must be synthesized or strategies modified to obtain superior and/or modern photocatalytic performance in the removal of contaminants. Modified hydroxyapatite (HAp) photocatalysts have gradually emerged in photocatalysis as a series of promising compounds, bringing new opportunities in pollutant remediation [22–25]. These photocatalysts have gained prominence in the photooxidation of contaminants due to biocompatibility, bioactivity, nontoxicity, and low cost [26,27], which has been widely reported in adsorption applications [27–30].

This review is a brief, critical, and comprehensive prospection about photocatalysis for treating water contaminated by dyes, drugs, and pesticides, using HAp for the remediation of pollutants. The topics were (a) principles of heterogeneous photocatalysis combined with parameters that affect degradation; (b) properties of HAp-based photocatalysts; (c) strategies and roles (doping, support, substrate, and co-catalyst) of HAp; (d) the synthesis processes; (e) photodegradation of organic contaminants in water, especially dyes, pharmaceuticals, and pesticides using HAp; and (f) the challenges of and perspectives on the development of these compounds for application in the field.

The main idea of the study is to investigate the impacts of various process parameters of hydroxyapatite-based materials on the degradation of organic contaminants. Only articles published between 2011 and 2021 were examined for the advanced bibliographic search. The databases of articles—Web of Science (Topic-TS), PubMed (All Fields), and Science Direct (Find articles with these terms)—were employed. For patents, the European Patent Office (EPO), United States Patent and Trademark Office (USPTO), and Brazilian National Institute of Industrial Property (INPI) were used as search mechanisms. For the word "contaminant degradation" (the main topic), the following refinement keywords were delimited: calcium phosphate; hydroxyapatite; hydroxyapatite or HAp; photocatalytic degradation; photocatalytic properties; hydroxyapatite and photocatalytic degradation; hydroxyapatite and photocatalytic properties; hydroxyapatite and drug degradation; hydroxyapatite and dye degradation; and hydroxyapatite and pesticide degradation. Searches in search fields in article databases were restricted in advanced searches, and review articles were excluded. In the patents, the search elements were abstract (USPTO), with the compound words being limited by quotation marks, such as "calcium phosphate", "photocatalytic degradation", and "photocatalytic properties." The title and abstract were the connectors for the EPO data and the abstract for INPI.

The first five keywords brought up many publications because these expressions cover several fields and areas of study. The quantity of articles found for the terms "hydroxyapatite and photocatalytic properties", "hydroxyapatite and dye degradation", and "hydroxyapatite and pesticide degradation" were considerably lower than the others, indicating the possibility for future studies on the proposed theme. Due to the extensive number of articles found in the databases, we restricted the dates, search modes and topic connectors, abstracts, and all fields used. Table 1 shows the number of publications found in the databases mentioned above.

**Table 1.** Number of articles found in the Web of Science, Scopus, and SciELO databases.

| Keywords | Number of Articles Found in the Databases | | |
| --- | --- | --- | --- |
| | **Web of Science** | **PubMed** | **Science Direct** |
| | **2011–2021** | **2011–2021** | **2011–2021** |
| Calcium phosphate | 26,863 | 39,555 | 111,528 |
| Hydroxyapatite | 30,695 | 36,415 | 26,794 |
| Hydroxyapatite or HAp | 33,146 | 44,605 | 5029 |
| Photocatalytic degradation | 60,415 | 12,613 | 42,285 |
| Photocatalytic properties | 40,039 | 18,524 | 52,894 |
| Hydroxyapatite and photocatalytic degradation | 142 | 961 | 390 |
| Hydroxyapatite and photocatalytic properties | 99 | 1252 | 575 |
| Hydroxyapatite and drug degradation | 287 | 12,671 | 3782 |
| Hydroxyapatite and dye degradation | 91 | 4248 | 2016 |
| Hydroxyapatite and pesticide degradation | 5 | 408 | 195 |

Several articles have used silver compounds to support different calcium phosphates [31,32]. HAp, considered a phosphate in combination with other compounds and metals, has achieved positive results in photocatalytic performance and efficiency in antibacterial treatment [33–35]. Piccirillo et al. [36] used the compound $Ag_3PO_4$ combined with HAp to analyze the antibacterial activity under UV and white light irradiation. The authors found that the antibacterial property of the synthesized material resulted in 99% inactivation percentages against *Staphylococcus aureus*, *Escherichia coli*, and *Pseudomonas aeruginosa bacteria*, thus demonstrating great potential for this compound. Furthermore, photocatalytic tests achieved 60% degradation of the methylene blue dye under UV radiation [37].

Another category is the use of HAp as an adsorbent to remove various substances, including protein molecules, as reported by Hirakura et al. [38]. The characteristic of the fibrous structure and the void spaces contribute to the adsorption and subsequent photocatalytic decomposition [39]. Another application of HAp is for polymeric films [40]. For example, Chu et al. [40] investigated calcium phosphate and $TiO_2$ to create a multilayer structure with silicone polyester to degrade the methylene blue dye. They concluded that the higher the molecular weight of the polyesters, the more stable and resistant the molecular structures are to decomposition by photocatalysts [40].

Overall, hydroxyapatite is presented based on parameters from different approaches and applications, for example, HAp adsorption for organic pollutant removal [39,41–44]; quantitative analysis for pollutant detection [45,46]; pollutant removal by filtration and precipitation [47,48]; medical applications [49,50], antimicrobial properties [51,52]; and application to photocatalytic degradation [53–57].

The research results carried out in the patent databases are presented in Table 2. The numbers were found for the terms *hydroxyapatite and "drug degradation", hydroxyapatite and "dye degradation" and hydroxyapatite and "pesticide degradation.* This shows that research on this topic is innovative, corroborating the low amount of research indicated in the articles. In addition, the American and Brazilian patent databases did not contain data on deposits for the terms mentioned; therefore, the work conducted in this area can provide significant support for national technological development.

**Table 2.** Number of patents found in the USPTO, EPO, and INPI patent databases.

| Keywords | USPTO | EPO | INPI (Portuguese) |
|---|---|---|---|
| Calcium phosphate | 1195 | More than 10,000 | 84 |
| Hydroxyapatite | 669 | 9350 | 56 |
| Hydroxyapatite or HAp | 4779 | 9882 | 0 |
| Photocatalytic degradation | 17 | 3688 | 1 |
| Photocatalytic properties | 21 | 1284 | 3 |
| Hydroxyapatite and photocatalytic degradation | 0 | 4 | 0 |
| Hydroxyapatite and photocatalytic properties | 2 | 1 | 0 |
| Hydroxyapatite and drug degradation | 0 | 11 | 0 |
| Hydroxyapatite and dye degradation | 0 | 1 | 0 |
| Hydroxyapatite and pesticide degradation | 0 | 4 | 0 |

In the analysis of the 16 patent deposits in the EPO (11 being for the term hydroxyapatite and "drug degradation", one patent for the expression hydroxyapatite and "dye degradation", and four patents for the term hydroxyapatite and "pesticide degradation"), two similar patents appeared among the expressions, leaving 15 inventions for study. Although the search presented keywords related to the review, a more detailed search showed that there is no emphasis on the water treatment process or its use as a photocatalyst. The lack or low number of patents on HAp in photocatalysis may indicate that this area of research is promising and that there is still no direct relationship between knowledge

producers (industries, companies, research centers) and universities, as a center of applied science and technology [58,59].

## 2. Literature Review

*Fundamentals of Heterogeneous Photocatalysis*

Heterogeneous photocatalysis is a promising advanced oxidation process (AOP) for mitigation and environmental control, as an ecologically capable tool in mild operating conditions compared to conventional processes for the biodegradation of numerous pollutants [60,61]. This technology is based mainly on semiconductor materials, such as chalcogenides, metal oxides, and mixed and/or binary oxides, which are considered essential photocatalytic materials [62,63].

A photocatalyst is considered a substance that can produce chemical transformations of the reaction partners under light absorption (solar, UV, visible) [61,64]. Furthermore, heterogeneous photocatalysts are semiconductors that act catalytically with a crystalline structure and a bandgap suitable for generating photo emitted charge carrier pairs (electron/holes), which are responsible for the oxidation-reduction reactions that ensure the photocatalytic cycle [65]. Similarly, the ability of photoinduced charge carriers to react with the adsorbed species on the surface of the photocatalyst depends on the energy positions of the semiconductor band (HOMO/LUMO) and the redox potential of the adsorbates that determine the possibility of the photocatalyst to promote oxidation reduction reactions [18,66].

The main advantages of this technology include the ability to use solar energy; the possibility of degrading and mineralizing into harmless products, generating secondary waste at a minimum scale; and the ability to recover/reuse after the process [67,68].

Heterogeneous photocatalysis is a photochemical process that involves a semiconductor under natural or artificial light. The classical photocatalytic degradation mechanism usually involves a series of redox reactions on the surface of a semiconductor [69–71], whose schematic representation can be seen in Figure 2.

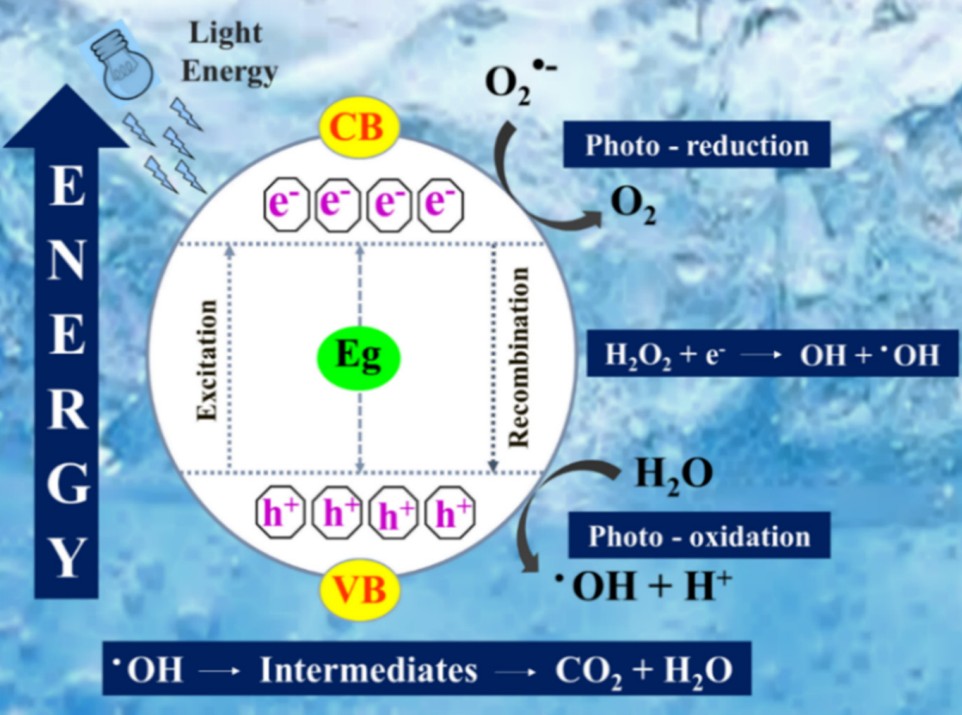

**Figure 2.** Schematic of the photocatalytic mechanism using a semiconductor.

The heterogeneous photocatalytic mechanism is initiated through a semiconductor that absorbs photons (*hv*) of energy equal to or greater than the bandgap (*Eg*), generating an excitation of electrons from the valence band (VB) to the conduction band (CB), forming electron pairs—holes ($e^-/h^+$). Subsequently, these photogenerated charge carriers can recombine or migrate to the surface of the photocatalyst [58,72,73]. At this time, charge carriers can directly participate in oxidation-reduction reactions that subsequently involve free radical species (especially hydroxyl radicals) and attack most organic pollutants [74,75] to break down and/or degrade smaller fragments or byproducts, such as carbon dioxide, water, and inorganic ions [76]. Based on semiconductor photoactivation and photogeneration and/or separation of $e^-/h^+$ pairs to form hydroxyl radicals and intermediates, we summarize here the possible oxidation-reduction reactions [77–79].

According to Mestre et al. [80], the relationship between the fast kinetic competition of recombination and the slow kinetics of separation and charge transfer (i.e., migration) to the material surface is significant during the process, as it determines the photocatalytic efficiency. Therefore, the first step is to try to avoid the recombination process (the primary limitation) to increase the photo-oxidation of the target contaminant.

Qian et al. [69] stated that two essential pathways can explain photocatalytic efficiency: (1) the competition between the recombination of charge carriers and (2) the entrapment and charge transfer.

Different works [81,82] report that such degradation processes comprise two stages of well-defined mechanisms: direct and indirect. These mechanisms are further described by Ajmal et al. [83], Kumar et al. [84], Natarajan et al. [85], Honorio et al. [72], and others [18,86].

In the photocatalytic process, operational parameters affect the reaction performance [87]. The initial pollutant concentration; photocatalyst concentration; solution pH; oxidizing agent type; calcination temperature; light intensity and type; photocatalyst structure and shape; doping or substrate type; and reactors are all relevant factors for its use, as it minimizes costs and process optimization [88,89].

The effectiveness of photocatalysts is associated with the operating conditions and surface properties of the materials. Therefore, they must have an adequate bandgap, surface area, particle size distribution, crystal structure composition, stability with respect to photocorrosion, nontoxic nature, low cost, and physical characteristics that allow them to act as photocatalysts [68,90].

Research has been conducted using a series of materials, such as $TiO_2$, ZnO, $ZrO_2$, CdS, g-$C_3N_4$, $Fe_2O_3$, and $WO_3$, and their various combinations have been examined as photocatalysts to degrade organic contaminants [89,91,92]. For example, titanium dioxide ($TiO_2$) is used as a reference photocatalyst due to its non-toxicity, water insolubility, hydrophilicity, adequate bandgap, and good chemical stability [60,93,94]. However, it has some limitations that are circumvented based on electronic band structure modification, including doping, heterojunction, immobilization, and sensitization strategies [95–98].

To optimize photocatalysts, synthesizing new compounds with specific properties and combating energy and environmental problems are promising methods to overcome the mentioned disadvantages and challenges for environmental impacts [80,99,100]. The classes of alternative compounds include ferrites [101,102], zeolites [103,104], niobates [105,106], stannates [107–110], clay materials [73,111–115], carbon materials [116,117], double hydroxides [118,119], polysaccharides [95,120], and graphenes [121,122]. Another alternative is the combination of substrates–oxide–metal–supports with hydroxyapatite photodegradation of a large number of pollutants, which will be given a more specific approach in this review.

## 3. Hydroxyapatite-Based Photocatalyst

Calcium phosphates are interesting compounds in an interdisciplinary field involving chemistry, biology, medicine, and geology. The first attempts to determine their composition by chemical analysis began in the first half of the 18th century by Berzelius [123]. A century later, Hausen introduced different phases of calcium phosphate crystals, and their mixtures were called apatites [124]. The apatite group comprises isomorphic hexagonal minerals

and can be commonly found in nature as geological and biomineral materials in calcified tissues. Apatite usually has the formula $M_{10}(TO_4)_6X_2$, with a hexagonal symmetry (space group $P6_3/_m$), where M is a divalent cation ($Ca^{2+}$, $Sr^{2+}$, $Pb^{2+}$, others); $TO_4$ is a trivalent anionic group ($PO_4^{3-}$, $SiO_4^{3-}$, $VO_4^{3-}$, others); and X is usually a monovalent anion ($F^-$, $OH^-$, others) [125–127].

Calcium hydroxyapatite (HAp) is a calcium phosphate of the formula $Ca_{10}(PO_4)_6(OH)_2$ with a composition and stoichiometric ratio of 1.67 (Ca/P), similar to bone apatite [44,128–131]. HAp can have two possible structures: monoclinic and hexagonal (space groups P21/b and P63/m, respectively), the most common form being hexagonal, in which calcium has two possible positions: Ca (I) and Ca (II) [132–135]. According to Boanini et al. [134], the structure can be described by a set of phosphate groups crossed by parallel channels filled by $OH^-$ ions and parallel to the crystallographic axis (c). Its structure allows other groups to replace $Ca^{2+}$, $PO_4^{3-}$, and $OH^-$ ions that can affect network parameters, crystallinity, crystal dimensions, surface texture, solubility, spectral properties, and thermal stability [134,136].

HAp can be derived from natural and synthetic sources [137], present in igneous rocks, sediments, soil, and suspended particles. It is a major component of bones and teeth [138–140]. HAp has numerous exciting properties, such as biocompatibility, high stability, and low economic cost due to its high availability, making it an attractive bioceramic [139,141]. A unique attribute is that it accepts many anionic and cationic substances due to its accommodation structure. Furthermore, its biomedical advantages and mechanical properties have made HAp a prominent compound in many applications [138,142–145]. For example, due to the similarity of HAp in the chemical composition of the mineral phase of bone tissues, it is known for its medical applications in tissue engineering [138,146]. In the pharmaceutical industry, HAp has demonstrated the ability to increase drug loading homogeneously, in addition to regulating its release by pH [44,139,147]. In agriculture, HAp was efficient in the controlled release of urea as a nanofertilizer [148]. Moreover, there is interest in its application in aquatic chromatography and remediation [149].

Regarding photocatalytic application, alternative strategies are evidence for the use and catalytic behavior, with functionalized materials being used to replace pure HAp and directed to catalytic applications and environmental protection [150,151].

Brazón et al. [146] report that HAp has photocatalytic properties, which are even more evident when combined with oxides to ensure significant effects due to the interaction and synergistic effect between the components.

To explain the photocatalytic behavior of HAp, Section 4.2 summarizes characteristics that demonstrate the fundamental role of HAp from different synthesis methods. Plausible considerations are based on the production of oxidizing species for the study model pollutant, in addition to reinforcing the enhanced activity of these compounds through modifications [35,152,153].

### 3.1. Synthesis Methods

Hydroxyapatite has been studied for decades due to its diversity and applications in environmental and biomedical management [154]. It has been used in various forms and/or nanostructures, including powders; granules; films; porous blocks; dumbbells; leaves; flowers; sticks; plates; and irregular, cylindrical, and spherical tubes [137,155,156]. In addition, a variety of processes and methods for synthesizing HAp and modified derivatives use different and reagents [137,151,155–159] (Figure 3). The synthetic methods or techniques (sol–gel, solid-state, mechanochemical, hydrothermal [156], precipitation, hydrolysis, combustion, and ultrasound) can be classified by physical and chemical methods [137,155,158–160].

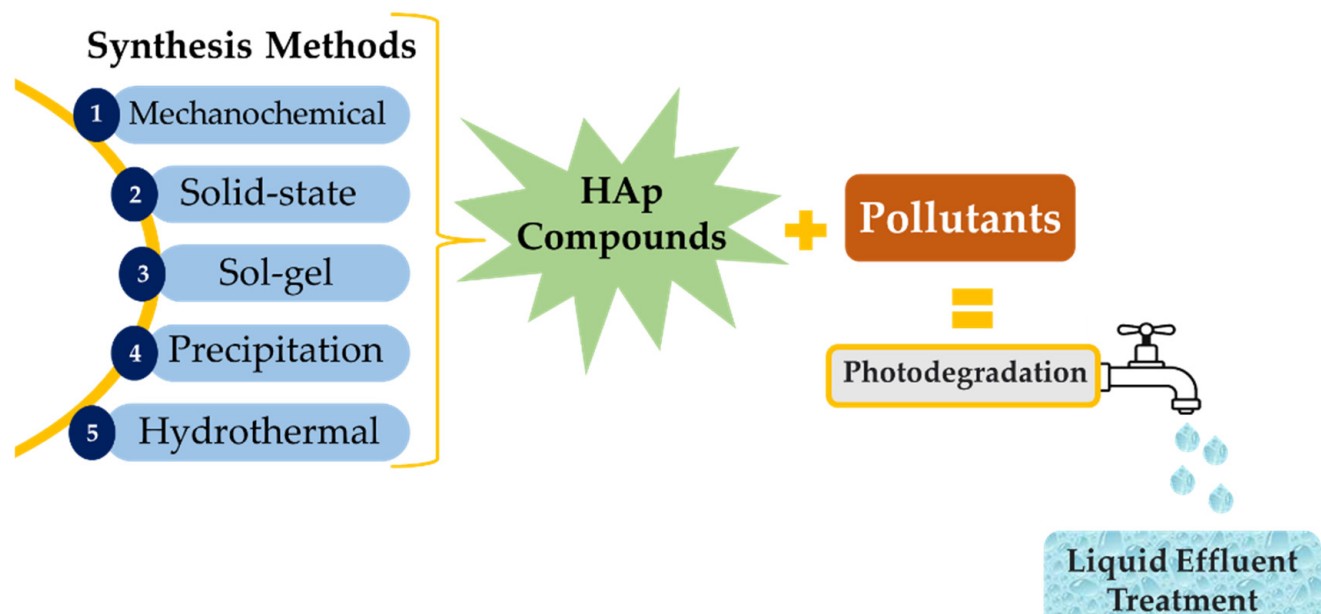

**Figure 3.** Summary of various synthesis methods of HAp applied to photodegradation of pollutants.

Each method involves several types of chemical synthesis routes and processing conditions according to the classification of advantages and disadvantages. Due to the wide variety available, choosing a specific route to synthesize well-defined HAp for a specific application can be laborious [137]. Therefore, in this review, we define five synthetic methods that are widely used to obtain HAp-based compounds.

- **The solid-state method** is a relatively simple procedure based on solid-ion diffusion from solid chemical precursors and subjected to high temperatures to produce new solids, especially the powder form. In the case of HAp, precursors are chemicals that contain calcium and phosphate [155,157]. The advantage is its low cost, despite requiring high calcination temperatures. However, its disadvantage is that this method generally leads to low phase purity, i.e., exhibits heterogeneity in its phase composition [137,155,158,160].

- **The mechanochemical method** applies compression, shear, or friction through the grinding process to produce a chemical transformation [157,161]. According to Agbeboh et al. [137], in this procedure, particle size decreases with increasing duration of mechanical activation. The advantage is a room temperature process. The disadvantage is low phase purity [137,155,158,160].

- **The sol–gel method** is used to obtain HAp with different particle morphologies. It is defined as the method based on forming a colloidal sol, which will later turn into a gel [158,162]. This method can produce excellent dimensions; however, the disadvantage is also variable phase purity [137,155,158,160].

- **The hydrothermal method** uses the solution hydrothermally treated at high temperatures and pressure in an autoclave or microwave. The pressure and temperature increase reactivity, and the effect of condensation facilitates chemical bonds forming nuclei that can obtain the compound with defined stoichiometry and crystalline structure [137,158]. It produces excellent mechanical and morphological properties. However, this method requires high pressures and temperatures above room temperature [137,155,158,160].

Due to the complexity in producing the desired property, such as phase purity, crystallinity, stoichiometry, size, and morphology, Table 3 presents numerous research and synthesis details in preparing HAp and derivatives applied to degradation.

**Table 3.** Summary of the synthesis methods and experimental parameters using HAp-based photocalystic sterilization.

| Material | Targeted Pollutant | Method Details | Synthesis Details and/or Precursors | T (°C)-t (h) | Ref. |
|---|---|---|---|---|---|
| TiO$_2$/HAp | Ciprofloxacin and ofloxacin | Dissolution/reprecipitation sol–gel | Nitric acid/Ti(OC$_3$H$_7$)$_4$/ammonia | 100 °C-24 h<br>500 °C-3 h | [163] |
| HAp<br>HAp-nTiO$_2$ | Methyl violet | Phosphatation of calcined cockle shells followed by dispersion and precipitacion | CaO/diammonium hydrogen phosphate | 1000 °C-3 h<br>400 °C-2 h | [164] |
| TiO$_2$/HAp | Methyl orange | Chemistry method | Tetrabutyltitanate/ethanol/HAp/water | 700 °C-6 h | [165] |
| ZnO-HAp | Paracetamol and Methylene Blue | Alkaline precipitation | Zn(NO$_3$)$_2$.2H$_2$O/NH$_4$OH (25%)<br>Zn$^{2+}$ ions/Ca$^{2+}$ ions and H$_3$PO$_4$ | 100 °C-24 h<br>500 °C-3 h. | [166] |
| HAp | Methylene blue | Wet precipitation | Calcium hydroxide/potassium dihydrogen phosphate/H$_2$O | 800 °C-5 h | [144] |
| Au loaded HAp | Methylene blue | Wet precipitation | Ca(NO$_3$)$_2$·4H$_2$O/(NH$_4$)$_2$HPO$_4$/NH$_4$OH | 600 °C-1 h | [153] |
| WMPHAP-Ti | Congo red | Chemisorption calcination | Titanium hydroxide/titanium isopropoxide/ethyl alcohol | 105 °C-12 h<br>700 °C-6 h | [167] |
| FB-HAp | Crystal violet and Congo red | Calcination methods | HAp/Catlabone (*Gibelioncatla*) | 900 °C-4 h | [168] |
| Oxide/HAp<br>ZnHAp/<br>TiHAp and FeHAp | Ciprofloxacin in | Dissolution/reprecipitation | Bidistilled acidified water/HNO$_3$ acid/ ammonia/tetraisopropyl orthotitanate, zinc, or iron nitrate | 100 °C-24 h<br>500 °C-3 h | [169] |
| CdS/HAp | Tetracycline | Low temperature Hydrothermal | Commercial CdS/HAp microspheres<br>Distilled water | 80 °C-4 h (autoclave)<br>60 °C-12 h (yellow product) | [24] |
| nHAp | Simulated body fluid | Atrazine | (1) Calcination of cow bone<br>(2) Conversion of CaO to CaCl$_2$<br>(3) Preparation of simulated body fluid | 900 °C-8 h<br>150 °C-3 h<br>The SBF solution prepared was stored in the refrigerator at a temperature of 5 °C. | [170] |
| Ga-HAp | Methylene blue | Suspension–precipitation | Ca(OH)$_{2(s)}$ + 0.05Ga(NO$_3$)$_{3(aq)}$ + 6(NH$_4$)$_2$HPO$_{4(s)}$ $\rightarrow$ Ca$_{9.95}$Ga$_{0.05}$(PO$_4$)$_6$(OH)$_2$ ↓ + 18H$_2$O$_{(l)}$ + 12NH$_{3(g)}$ | 110 °C-12 h. | [57] |

### 3.2. Design and Strategies of Hydroxyapatite-Based Photocatalysts

HAp can be obtained by several synthetic methods for water treatment due to its promising properties. Considering its potential application, HAp is a critical non-metallic material that acts with other inorganic phases (photocatalysts) to promote the photocatalytic degradation of ecologically correct, safe, and nontoxic pollutants [168]. Although HAp compounds can assist with environmental protection progress, many challenges need to be overcome with structural modifications and appropriate interface to extensively meet the requirements for propionate photocatalysts. As pure HAp does not present a direct/positive action in photocatalysis, structural modifications are necessary to improve the essential attributes (bandgap, particle size, surface area) of these derived compounds [163,171].

One of the pioneering works on the photocatalytic behavior of HAp was published in 2002 and describes the UV light-induced photodegradation of methyl mercaptan [172]. Nishikawa and Omamiuda [172] stated that the photo-functional behavior of HAp might be related to the superoxide anion radical, allowing a strong connection with the formation of vacancies on the surface of HAp observed in electron spin resonance (ESR). The authors indicated that electron transfer must occur from the vacancy formed in the apatite structure to oxygenate in atmospheric air in HAp. This observation was identified in methyl mercaptan conversion using HAp 200 under UV irradiation for 60 min and effectively achieved 96% decomposition. Table 3 Shows a summary of the synthesis methods and experimental parameters using HAp-based photocatalyst.

Clearly, Bystrov et al. [173] emphasize that although hydroxyapatite provides exciting results in the photocatalytic field, the fundamental roles that affect structural characteristics and which properties give it a photoactive character in specific configurations and morphologies are still unknown. Therefore, they describe a modeling study based on density functional theory using the HAp and correlate the acceptable oxygen vacancies present in the HAp network and the bandgap to explain such performance.

According to Reddy et al. [138], the use of hydroxyapatite as a photocatalyst under UV irradiation is well understood. The authors state that the catalytic phenomenon in HAp is due to photoinduced electronic excitation attributed to oxygen vacancies that contribute to the formation of radicals that can oxidize pollutant molecules and react with water and other ions, which later, by successive reactions, form $^{\bullet}OH$ and the degradation of the pollutant in question.

HAp has been used in environmental remediation due to its peculiar properties, such as biocompatibility, stability, and high specific surface area, which makes it attractive from a catalytic point of view, as it exhibits the potential for immobilization on metals HAp stably, ensuring the concept of green photocatalysts in photocatalytic reactions [22,174,175]. In addition, it is considered an adequate co-adsorbent due to its transparency to UV radiation, adsorption character in different aqueous pollutants, and low solubility in water [176]. Therefore, different forms and derived compounds in pure, doped, supported, and heterostructured forms are cited in the literature as improvements in degradation processes [39,144,177–179].

The doping process is a practical approach to improve the photocatalytic activity of $TiO_2$, for example [180–183]. In general, dopants in its structure can change electronic levels and shift the absorption edge to the visible region. Thus, the photoresponse is improved by the emergence of new energy levels [176,184]. In the case of hydroxyapatite, the physical, chemical, and biological properties are controlled by its structure and crystalline composition. One of its main characteristics is the ability to be substituted by other metal ions for calcium ions, bringing new properties to apatite [127].

Liu et al. [179] explored $Fe^{3+}$-doped hydroxyapatite for the degradation of rhodamine B (RhB) under visible light. The material prepared was relatively stable during the recycling experiments. Pang et al. [22] successfully synthesized Co-HAp to activate peroxymonosulfate in the degradation of organic contaminants, with 93.3% of RhB degraded in 12 min, and the performance was even more evident in the degradation of other organic pollutants (Acid Orange 7, tetracycline, and levofloxacin). The authors explained this behavior based

on the Electron Paramagnetic Resonance (EPR) results and the suppression of radicals and nonradicals involved in the system, in which the $^1O_2$ species was extremely important in the degradation of the dye. In addition, other properties were relevant for high catalytic performance.

Mohseni-Salehiet al. [176] presented excellent results in the degradation of methylene blue using titania nanoparticles doped with different concentrations of cobalt and nickel and titania/Hap-doped nanocomposites for photodegradation tests.

In the condition of catalytic support, many compounds can provide redox potential electronic characteristics and promote a better metal–support interaction to ensure stability between the metal and the host environment [185]. Strategies for hydroxyapatite activation have been systematically introduced to increase catalytic activity [22].

The reduction of the methyl orange azo-dye is used as a model pollutant to evaluate the catalytic activity of Pd nanoparticles supported on HAp [186]. Zhang and Yates reinforce the importance of the synthesis process and the support–metal interaction as a significant answer for the performance and reduction in the dye rate [186]. The heterojunction and/or immobilization strategies were reported by Zhang et al. [187] and Sulaeman et al. [188], using HAp composites in the degradation of atrazine and rhodamine, respectively.

Recently, Neelgund et al. [56] reported HAp as an important aspirating compound for photocatalytic applications, but it has not been well explored due to its low catalytic efficiency. To improve this, they proposed fusion with another compound that can modify the structure and increase the photocatalytic response, reducing its bandgap and inhibiting the recombination between the charge carriers.

Combining photocatalysts with complementary HAp within composite structures favors the dispersion of the photoactive phase, the increase in surface area and the formation and recovery of photoactive materials, thus improving intrinsic photocatalytic properties for multifunctional projection suitable for degrading pollutants [189]. Therefore, HAp seems particularly interesting due to its high metal ion retention capacity and good properties as a photocatalyst [132,167,190–192].

## 4. Photodegradation of Organic Pollutants in Water

Numerous hazardous environmental pollutants in both organic and inorganic water have become a global problem [193]. The threat of these contaminants and their complex mixtures are focused on the origin, occurrence, destination, transport, toxicity, and recovery, and it resides in the risks and undesirable effects for aquatic and terrestrial organisms due to the high biological activity, extensive treatment deficit, and difficulty in recovery because of the degree of nature vs. risks vs. concentration [7,39,194–196]. Therefore, combating organic pollutants is an important topic. This review focuses mainly on the degradation of dyes, drugs, and pesticides (pollutants most used in the degradation) by HAp-based photocatalysts in remediation (Figure 4). The following sections summarize the definitions and research related to each class of pollutant. Figure 5 shows the annual growth of the term "hydroxyapatite and photocatalytic application" according to the Science Direct database (advanced search), totaling 1540 articles published in the last ten years (including all types of documents), with 171 published in 2022. Although the mechanistic aspects of HAp are still minimally explored, its expansion has sustained optimistic projections in several areas of concentration, means of dissemination, and subject areas for environmental protection.

### 4.1. Degradation of Dyes

It is estimated that 700,000 tons of various dyes are sold, with 100,000 dyes manufactured each year [197,198]. Generally, after these have accomplished their purpose, the vast majority are discarded into the environment in significant amounts without prior treatment [197]. The presence of dyes in water is aesthetically undesirable and hinders the photosynthesis process, thus preventing sunlight penetration into the water and, consequently, affecting the aquatic life of plants, animals, and human health due to its toxic nature [199–201].

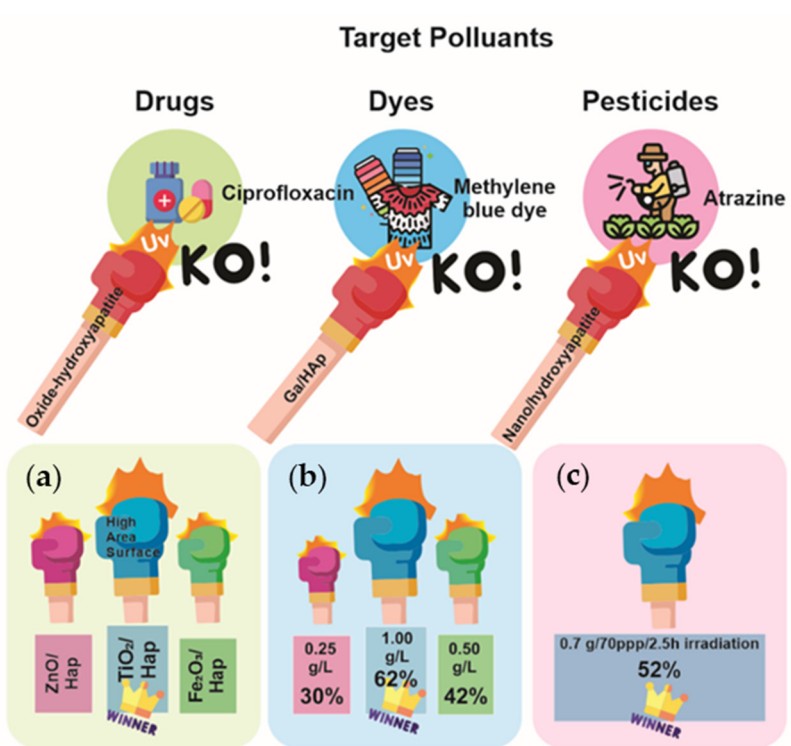

**Figure 4.** Details of the operational parameters of HAp oxide-based compounds applied to combat pollutants via heterogeneous photocatalysis according to the literature [57,169,170]: (**a**) different HAp-oxide based compounds for drug degradation; (**b**) different concentrations of HAp-Ga for dye degradation and degradation rate, respectively; and (**c**) nano-hydroxyapatite as photocatalyst showing 52% of atrazine degradation rate.

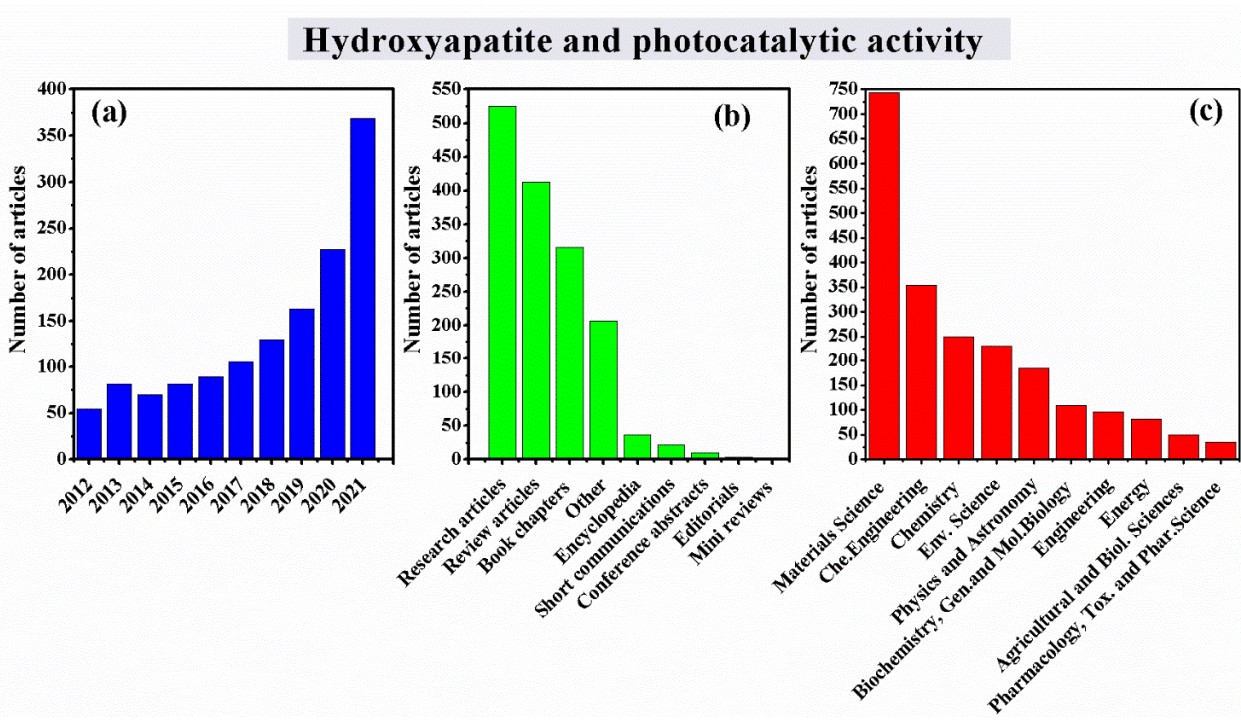

**Figure 5.** (**a**) Annual evolution of the number of articles according to the Science Direct database in the last ten years, using the term "hydroxyapatite and photocatalytic activity" (Advanced search— Research articles); (**b**) article type; and (**c**) subject areas.

Dyes are unsaturated organic compounds that absorb light and provide color in the visible region (380 to 750 nm). These absorb part of the visible spectrum (chromophore), with the color represented by the light fraction and intimately linked to their chemical composition [202–204]. In the conception of Katheresan et al. [197], dyes are colored substances designed to give color to materials, such as fabrics, papers, or any other material that can be colored.

Samsami et al. [205] state that textile industries constitute more than half of the effluents represented by dyes in the world aquatic environment (54%), followed by dye production (21%), paper and cellulose (10%), inks (8%), and dye production industries (7%). These percentages were also described by De Gisi et al. [206].

The constituent molecules of dyes are based on three main groups: the chromophore (responsible for absorbing light), the auxochromes (a water-soluble binding compound), and the matrix [203,207]. Chromophore groups consist of sets of atoms, such as nitro ($-NO_2$), azo ($-N=N-$), nitroso ($-N=O$), carbonyl ($-C=O$), and alkenes ($-C=C-$) [197,203,208] and are classified according to their molecular structure, color, and application, as mordant (Mordant Black 11), reactive (Reactive Blue 5), sulfur (Sulphur Black 1), vat (Vat Blue 4), acid (Acid Yellow 36), basic (methylene blue), azo (bluish-red azo dye), direct (Congo red), and reactive (Reactive Blue 5) [197,203,205,208].

Among the distribution of dyes in water, azo dyes are widely used in industrial processes and represent more than 70% of the world's annual production of synthetic dyes [198,209]. They are compounds defined by the presence of a double bond between two nitrogen atoms ($-N=N-$) bonded to aromatic rings [198].

Examples of azo dyes were studied by Sathiyavimal et al. [168] who described the synthesis of HAp from fish bones for environmental application (FB-HAp). In this study, the potential of FB-HAp was verified against the degradation of Congo red and crystal violet dyes, exhibiting 77% and 87% degradation, respectively, in 75 min of exposure. The degradation of azo dyes happened based on the proposed mechanism. The results were measured by the valence bond change to the conducting band that produces electron-hole pairs due to the irradiation of sunlight [168]. HAp and its compounds are environmentally-friendly and stable catalysts, and textile effluents are often very complex and require intensive treatment, such as photocatalysis, as an example. Table 4 presents works on the exclusive degradation of dyes.

### 4.2. Pharmaceutical Products or Drugs

Among the water pollutants that mainly affect water resources, drugs or pharmaceutical products have been identified as bioactive chemical products, and their contamination is evidenced by the growth in the production of these substances. However, they are not yet fully regulated and thus classified as emerging pollutants [216,217].

Medications are commonly found in many types of environmental matrixes, including surface, potable, and underground water. They are erroneously distributed in water treatment plants without specific pretreatment, causing adverse effects and even unknown potential in the aquatic system because they bioaccumulate and pose threats at all levels of the biological hierarchy [218]. Therefore, their elimination becomes a challenging task [217,219–221]. Furthermore, although their concentrations in aquatic environments are deficient, they can significantly impact aquatic and terrestrial ecosystems [54,222].

According to Kumar et al. [217] the term "pharmaceutical products" refers to a broad and diverse class of medications, including anti-inflammatories, analgesics, antidepressants, anxiolytics, antibiotics, antiepileptics, beta-blockers, synthetic hormones, antiepileptics, and others which have been detected in aquatic ecosystems. Furthermore, numerous other classes of pharmaceutical compounds have been reported in the literature [223–225].

Bekkali et al. removed pharmaceutical compounds ciprofloxacin (CIP) and ofloxacin (OFL) under UV irradiation [189]. The nanocomposites exhibited higher rates in the presence of high Zn charges in the composite structure than the isolated oxide due to better dispersion of the ZnO nanocrystals.

**Table 4.** Published articles in the field of wastewater dye removal using HAp compounds.

| Compound | Pollutant Target | Operating Parameters | Degradation of Results | Ref. |
|---|---|---|---|---|
| $TiO_2$/HAp | Methyl orange (MO) | Irradiation (300 W)<br>The dosage is 1 g/L–5 mg/L MO | $TiO_2$/HAp with a Ti content of 2.55% was 10 times higher than HAp. | [165] |
| HAp | Methylene blue | UV lamp–$O_2$ sparge-2.0 g/L MB | 54%-6 h/62%-24 h (Presence of $O_2$) | [144] |
| Pd/HAp/$Fe_3O_4$ | Methyl red,<br>methyl orange, and methyl yellow | Photocatalyst dosage: 0.1, 3.5, and 10 mg.<br>Fixed dye concentration: 5 $\mu$g ml$^{-1}$<br>pH: 1, 2, 7, and 10. | The photocatalyst activity is demonstrated by the study of first-order kinetics based on kinetic constants. | [145] |
| $Ag_3PO_4$/HAp | Rhodamine B | Under visible-light irradiation | 98.609%/4 h undred visible-light conditions | [210] |
| HAp@$TiO_2$ | Rhodamine B | 20 mg of powder—5 mg/L<br>Rh-B-UV (100 W) | HAp@$TiO_2$-700 exhibited the maximum catalytic activity among all the uncalcined and calcined nanocomposite samples, in which seven were higher than that of pure $TiO_2$ calcined at 700 °C. | [211] |
| $Fe_3O_4$@HAp | Acid Red 73 (AR73) | Photocatalyst dosage: 0.05–2.0 g/L<br>UV lamp—125 W (9.3 mW/cm$^2$)<br>Conc. AR73: 10 mg/L | Removal of AR73 de 97%.<br>The interaction of HAp and UV light activated photo-induced electronics, resulting in the formation of a vacancy on HAp by changing the electrons of the surface $PO_4^{3-}$ group. | [152] |
| HAp | Calmagite (azo dye) | Mercury lamp (250 W)<br>Conc. dye: 50 ppm<br>Hydroxyapatite: 3 g L$^{-1}$ | At optimum conditions (pH 6.5 and catalyst-3 g L$^{-1}$) presented complete degradation of calmagite (50 ppm) at 12 h, and a maximum of 92% COD removal was achieved with an increase in biodegradability of 0.78. | [138] |
| (HAp) nano-rods | Rhodamine | 0.15 g of Hapcatalyst to 50 mL (5 ppm) RhB | The potential application as a catalyst for photocatalytic degradation reached 86.8% degradation after 300 min under UV radiation. | [212] |
| HAp/Ta core–shell | Turquoise Blue GL | Photocatalyst: 25 mg<br>Conc. dye: 15 mg L$^{-1}$<br>UV light ($\lambda_{max}$ = 365 nm) | HAp/Ta core–shell NRs achieved 64.51% degradation at 0 h. Upon increasing the time to 8 h, the degradation increased to 98.63%. | [178] |
| CuHAp | Naphthol Blue Black | CuHAp: 0.05 g-Conc. dye: 5 mg/L | CuHAp showed a high catalytic effect of up to 70%. | [213] |
| ZnO-HAp | Rhodamine B and caffeine | (CAF) = 40 mg L$^{-1}$ and (RhB) 20 mg L$^{-1}$<br>0.1 g of w ZnO-HAP | The two pollutants using the 50ZnO HAp catalyst reached 90% for RhB and 65% for CAF under UV light. | [214] |
| Nano-HAp | Remazol Brilliant Blue R (RBBR) | RBBR: (10, 30, and 60 mg L$^{-1}$)<br>Photochemical UV reactor | Photocatalytic degradation of RBBR was ~80% achieved in 120 min using nano-HAp. | [215] |
| Au loaded HAp | Methylene blue (MB) | Under visible-light irradiation<br>40 mg of sample-Dye (5 ppm) | The total MB removal efficiency of the HAp nanoparticles increased to 32.47% at 0.055% by weight Au loading. | [153] |

Zou et al. [53] studied RGO/HAp composites prepared by the hydrothermal method and investigated the degradation of tetracycline (60 mg/L) under the presence of a 300 W xenon lamp. They observed that the compound RGO (1.5% by weight) exhibited a high performance, reaching 92.1% in 30 min of irradiation. Furthermore, RGO/Hap (1.5 wt%) exhibited considerable stability and repeatability, reinforcing its promising potential as an efficient photocatalyst.

The use of HAp composites decorated with small amounts of ultrafine graphitic carbon nitride ($gC_3N_4$) in tetracycline photodegradation (CT) was reported by Xu et al. [226]. The study showed that the composite produced with 1.5% of the mass of $g\text{-}C_3N_4$ had the highest photocatalytic activity in the degradation of tetracyclines (TCs) (almost 100% in 15 min). Moreover, this composite exhibited good stability in cyclic lanes for photocatalytic degradation of TCs. Finally, the study proposed the TC degradation mechanism through the $g\text{-}C_3N_4/HAp$ composite.

Within the class of antibiotics, tetracyclines (TCs) are categories that have a chemically stable structure, and low biodegradability is commonly prescribed to control/treat human and animal diseases, mainly due to antimicrobial effects [227]. TCs include the oxytetracycline, doxycycline, and chlortetracycline types, as shown by Minale et al. [228], Xu et al. [227], Jafari Ozumchelouei et al. [229], and Nguyen et al. [230].

*4.3. Degradation of Pesticides*

Pesticides are compounds used to exterminate, combat, or control pests [231]. In general, these are toxic organic pollutants with low biodegradability and are persistent in nature and have adverse implications for human and animal life if present above the acceptable concentration levels [232–235]. Furthermore, due to the long persistence periods and extensive use, these include numerous contamination vectors, hence the need for remediation technologies [236–240]. According to Vaya et al. [231], pesticides are classified in different ways (insecticide, herbicide, fungicide). This classification is also reported by Mojiri et al. [241]. Recent strategies simulate the degradation of persistent and toxic pesticides using HAp compounds and derivatives, as shown in Figure 6. Below are details of studies on pesticide control using hydroxyapatite-based compounds.

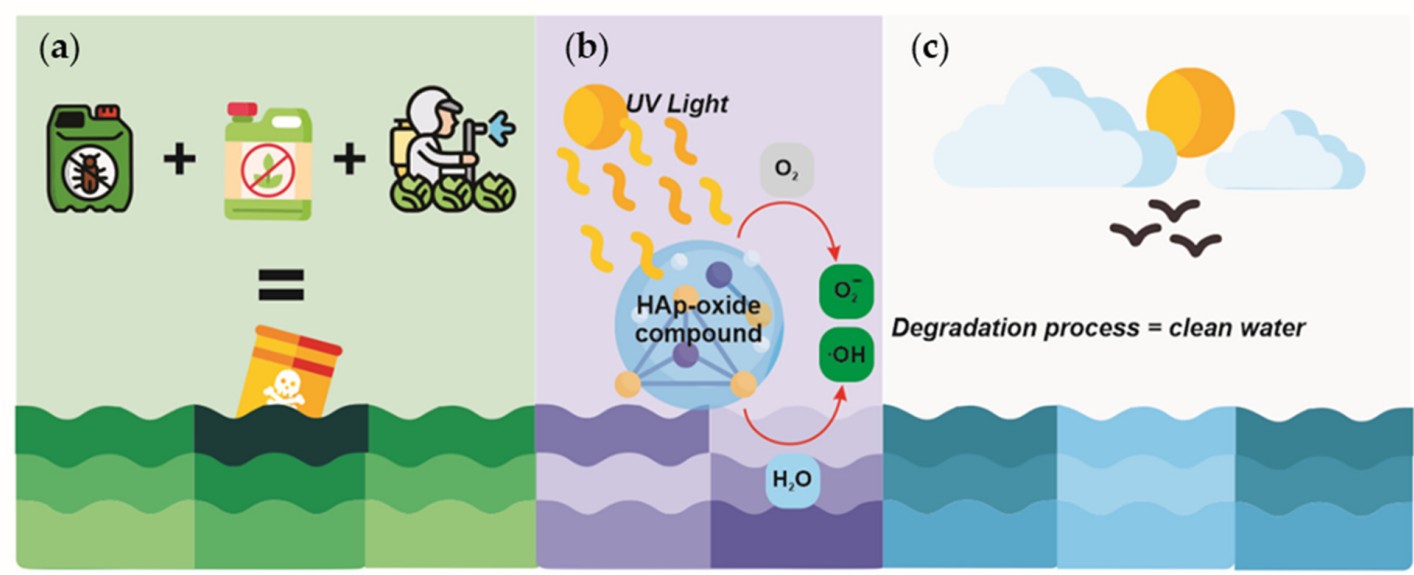

**Figure 6.** Simulation of pesticide degradation using HAp-oxide compounds and the purification process: (**a**) water polluted with pesticide, (**b**) main photocatalytic reactions using HAp-oxide compounds, and (**c**) clean water after the photocatalytic process.

Rubi et al. [170] investigated the use of nano-hydroxyapatite (HAp) in the degradation of atrazine (herbicide). The amount of photocatalyst dosage and the UV exposure time were substantial in the efficiency of the process, reaching a degradation of approximately 52% with 0.7 g of Nano-HAp for 2.5 h of irradiation. The results indicated that atrazine could be positively reduced without generating harmful by-products compared to the starting pollutant after the process.

Hoang et al. evaluated silver nanoparticles (AgNPs) immobilized on HAp and layered double hydroxides (LDH) in the catalytic reduction in 4-nitrophenol [25]. According to the authors, the sizeable superficial hydroxyl groups of HAp and LDH were exploited to form hydrogen bonds with carboxylate-capped AgNPs, favoring the degradation of 4-nitrophenol. The excellent performance was attributed to the proportionality of the silver content and the decrease in particle size, with the rate constants of the reduction in 4-nitrophenol catalyzed by Ag/LDH (4%–6% by weight of Ag), about 2.5 times greater than pure AgNPs. Thus, decorated AgNPs were confirmed as stable, ecological, reusable, and promising photocatalysts for catalytic applications. Another study using silver nanoparticles (AgNPs) and functionalized by dopamine was applied in the degradation of 4-nitrophenol(4-NP)[242]. The decorated compost was verified to be an efficient, ecological, and heterogeneous catalyst with potential for future industrial applications [242].

Flores et al. [243] correlated the photocatalytic activity of $Tb^{3+}$ doped hydroxyapatite for photodegradation of 2,4-dichlorophenoxyacetic acid under UV irradiation. The $^{\bullet}OH$ radicals were responsible for the degradation of (2,4-D) via the corresponding use of 10% by weight of terbium, achieving almost total degradation and 95% mineralization of the organic molecule after 240 min of irradiation.

The degradation of Connect pesticide was studied by Lindino et al. [136]. Using the transition metal-doped hydroxyapatite as a photocatalyst, 100% efficiency was achieved for W and V dopants under UV irradiation in 1 h. The excellent efficiency was attributed to the factors of crystallinity, surface area, pore size, adsorption capacity, and stability.

## 5. Challenges and Perspectives

The excellent characteristics of HAp are known in numerous sectors of science and research. The properties allow applications ranging from adsorbents to supports for various applications. For photocatalytic application, the use of HAp to obtain new photocatalysts through several sustainable synthesis processes that remove pollutants has progressed in recent years. The ability of HAp to generate new chemical compositions from the stoichiometric deficiency has been crucial in the derivation of HAp-based powders, films, and templates for sustainable, nontoxic, and low-cost composites based on green chemistry principles.

To improve the catalytic activity of the processes, the functionalization of the physical–chemical modification directly assists in the production of active sites, so HAp and derivatives could better mitigate environmental pollution, including the mineralization of pollutants into carbon dioxide ($CO_2$), water ($H_2O$), and inorganic ions ($SO_4^{2-}$, $NO_3^-$, $Cl^-$). However, to improve catalytic performances, many challenges remain for the use of HAp in the treatment of aquatic settings using photocatalysis as a remediation tool for factors such as the difficulty in recovering the medium, uniformly distributing light over the entire surface of the photocatalyst, catalytic loss over exposure time, and ease of sintering. In addition, the difficulty in studies about the catalytic mechanisms involved at the commercial level remains because the pilot/industrial-scale application is still limited in this area of knowledge, restricting viable proposals for commercial optimization to treat large amounts of effluent. Fihri et al. [133] explained that the textural properties and the mesoporosities are challenges to be explored, as are the weak mechanical properties that make long-term yields impossible due to the difficulty of control, mainly when HAp decomposes in an acidic medium [133].

Future efforts should focus on optimizing the catalytic activity and selectivity of the photocatalyst, avoiding the issue of poisoning and recyclability and overcoming the lack of

understanding about the catalytic mechanism. For all the points mentioned above, these approaches will probably allow significant advances in the environmental, agricultural, industrial, medical, and several other sectors that could use HAp as a good photo caliber, guaranteeing ecological improvements of an economic and social character, as well as the direct contribution of improved processes in environmental remediation.

## 6. Final Remarks

This review describes significant applications of HAp compounds and derivatives in the degradation of organic pollutants, emphasizing the removal of dyes, drugs, and pesticides in the studies about structural modifications and their properties, stability, and antibacterial activity. Current challenges and various scientific findings related to HAp and derivatives in aquatic treatment are concisely summarized. Overall, such compounds act as efficient photocatalysts due to their ease of removing contaminants from wastewater using advantageous redox activity and other properties already discussed here. The synergistic effect is supported as structural changes minimize ecological and environmental problems. This review emphasizes, discusses, and reports the state-of-the-art of these promising compounds for use and insights into their efficiency, ease of synthesis, and nature for removing pollutants from wastewater streams, mainly using heterogeneous photocatalysis as an active methodology in the destruction of wastewater. According to the knowledge acquired so far, in addition to the numerous applications of HAp in tissue engineering, doped, supported, incorporated, and heterostructured HAp were found in photodegradation studies. Therefore, considering the rapid advances in technology in discovering new compounds, many promising indicators can be applied to raise even more challenges in science and research.

**Author Contributions:** Conceptualization, P.T.; methodology, R.L.P.R.; writing—original draft preparation, R.L.P.R. and L.M.C.H.; writing—review and editing, P.T., T.M.D., R.D.d.S.B., M.G.F., E.C.S.-F. and J.A.O.; visualization, T.M.D., E.C.S.-F. and J.A.O.; supervision, J.A.O.; project administration, J.A.O. All authors have read and agreed to the published version of the manuscript.

**Funding:** This research was funded by CAPES, CNPq, and FAPEPI.

**Acknowledgments:** The authors would like to thank their institutions, UFPI, UFPB, and IFPI Institutes.

**Conflicts of Interest:** The authors declare no conflict of interest.

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
