# Peer review of "Light-Activated Hydroxyapatite Photocatalysts: New Environmentally-Friendly Materials to Mitigate Pollutants"

_minerals, doi:10.3390/min12050525_

Round 1

Reviewer 1 Report

Review

Title: Light activated hydroxyapatite composites: New environmentally benign compounds to mitigate pollutant

Authors: Rafael Lisandro P. Rocha, Luzia Maria C. Honorio, Roosevelt Delano de S. Bezerra, Pollyana A. Trigueiro, Thiago Marinho Duarte, Maria G. Fonseca, Edson C. Silva-Filho and Josy A. Osajima

In this review, the authors present a synthesis of the articles and patents reported for treatment of contaminated water using HAp-based compounds. Moreover, in this review are underlined the recent advances in the photocatalysis domain through HAp-derived photocatalysts.

In my opinion, this work it deserves to be published after the authors make the following minor changes:

  1. Some studies on the materials used in these applications have been omitted. Therefore, please see an refer the following papers:
  • https://doi.org/10.1016/j.clay.2016.08.019
  • DOI: 3390/ma11081350
  • https://doi.org/10.3390/polym13101617
  • https://doi.org/10.1063/1.5045983
  • https://doi.org/10.1155/2014/361061
  • https://doi.org/10.1155/2014/176426

Author Response

Dear Reviewer 1,

Thank you very much for your attention and the reviewers’ comments on our manuscript Light activated hydroxyapatite photocatalysts: New environmentally materials to mitigate pollutants(minerals-1634926)”. We agree with the comments and suggestions, which have been of great assistance in improving the quality of our paper and guiding our research. We have revised the paper after carefully studying the reviewer’s comments and have responded to them point by point. Revisions to the manuscript are highlighted in red. We have also made several further changes after carefully rereading the manuscript, which are likewise highlighted in red. We have tried our best to improve the manuscript. These changes do not influence the content and framework of the paper. We appreciate the editors’ and reviewers’ work, and hope that the revisions and accompanying responses will make our manuscript suitable for publication in this journal.  Once again, thank you very much for your comments and suggestions.

Yours sincerely, 

Dr  Josy Anteveli Osajima

Reviewer 2 Report

General comment

The review “Light activated hydroxyapatite composites: New environmentally benign compounds to mitigate pollutant” summarized the state of the art in terms of scientific papers and patents on the use of hydroxyapatite for photo-activated environmental remediation. The topic is interesting and well deserves a review article. The idea of studying also patents is good. However, the review has serious flaws as is poorly organized and often goes off topic, some key points are missing and other are over-redundant, and figures do not give any additional insight. In addition, there are several typos, grammar errors, misused words, changes in font size, or broken sentences that make more difficult to read and comprehend the text. The review can be improved after a significant structural reworking.

Specific comments

There are too many errors, repetitions, and other typos to list them all. Only the major flaws will be listed below.

Title

  • The title itself is misleading. In the review there are no “composites” (i.e. combination of two different materials). Also the part “New environmentally benign compounds to mitigate pollutant” is confusing: there are environmental malign anti-pollution agents? What does it mean?

Introduction

  • Some example of incorrect word use and/or non-understandable sentences:
    • the low demand for water resources (low?)
    • thousands of different molecules are released from polluting sources onto the market each year (there are polluting sources in the market?)
    • The vast majority of these compounds are toxic to microorganisms and, therefore, are not easily destroyed and eliminated by nature due to their high stability to light and temperature [1], requiring the use of remedial technologies capable of mineralizing these compounds considered non-biodegradable
    • To protect the aqueous environment from such toxic pollutants, various destructive and non-destructive classification methods, (classification?)
    • advanced oxidation processes (AOPs) …. By definition, POAs are defined by water treatment processes
    • Pubmed (All Filds)
    • How contaminant degradation is the main topic,
    • the possibility of degrading and mineralizing in harmless and/or hazardous products,
    • According to literary consolidation,
    • irradiated by natural or artificial photons (is not the photo, the light source can be natural or artificial)
    • Hydrothermal Method: Method that uses the solution hydrothermally treated (then the text is interrupted, and after there is a much longer text on hydrothermal methods)
    • Design and strategies of hidroxipatite-based photocatalysts
    • explained such behavior based on the EPR results (acronym not defined)
  • Hydroxyapatite is briefly mentioned in the introduction. Considering that is the topic of the review, it should be discussed in a paragraph later or it should be discussed in detail there. Why apatite is promising for photodegradation? What potentialities has/problem that are solved? Why apatite should be better than the other materials?
  • Figure 1. The years can be put directly in the x-axis instead of using letters and putting years in a small inset. The same applies for figure 6.

Research planning

  • According to the text, all searches have been made using the “hydroxyapatite” keyword only. This can be a problem, as a lot of research article use the term “apatite”, and these work would not appear if it is strictly searched “hydroxyapatite”
  • Maybe considering the scope of the work, the topic and article/patent search should not be limited to hydroxyapatite only, but also to other calcium phosphates. There is a reason why the search is limited to apatite only?
  • I do not understand the scope of the paragraphs “Screening of published articles” and “Screening of published patents”. What is the scope or insight? No further screening/result refinement has been done. After, there is a brief description which seems more suited to be in the main text of the review
  • The patents search and screening should be made more in deep and more accurately. Reading the titles of the patents in table 3, almost all look out of scope (clearly medical products) except 2 out of 15

Literature review

  • The paragraph 3.1 is too long for the scope of the review, and unclear. It should be rephrased to explain more clearly and in a shorter way what are the principles of photodegradations, and what are the requisites for a material for such application
  • Figure 3. Several terms are unclear. What is the difference between “precipitation” and “co-precipitation” as are used as synonyms? The same applies for “sonochemical” and “ultrasound”. In addition, “natural sources” is not a synthesis method.
  • In general, the 4.1 paragraph is not on topic and the scope of the review is not on apatite synthesis. Since there is a plethora of synthesis methods of HAp, how does this relate to photodegradation? What are the key points? What are the targets/desired requisites?
  • Paragraph 4.2 is not clear, maybe as it reflects the literature. What are the photo physical properties of apatite, and how they connect to photodegradation? What should be the ideal characteristics for this application?
  • The same mention of Pang et al and the use of CoHA is repeated two times in few paragraphs
  • Figure 4, Figure 5, figure 7 (and also figure 3). All these figures are generic schemes that do not facilitate or convey messages relevant to the review. In detail figure 3 is just a list of methods without context/comment, figure 4 is three names (and the arrows seem to indicate that there is a connection between dyes, pesticides, and pharmaceutical products?), figure 5 says that “main groups” (of what?) are herbicides/fungicides/insecticides that kill plants/fungi/insects (trivial), and figure 7 is another name list partially off topic with the scope of the review. I’d suggest to give more insights for these images (like putting the relevant pro/cons of the techniques, or examples of HAp that have been used for those pollutant) or, also, putting images from the most interesting articles on the topic showing their photodegradation behavior/morphology/etc.
  • The whole paragraph 5.4 is off topic. In the abstract and introduction it was stated that the scope of the review is the use of HAp for photodegradation of pollutants, so what is the connection with the antibacterial effect that some HAp have? There is no photodegradation, and bacteria are not pollutants. The paragraph could be in topic if it would be dedicated on the use of HAp and photodegradation to induce an antimicrobial effect.
  • The same problems apply partially to section 6. In addition, since there are no papers on these topics, these considerations are more suitable to be made more concise and on-topic in the introduction section.

Author Response

Dear Reviewer, 2, 

Thank you very much for your attention and the reviewers’ comments on our manuscript Light activated hydroxyapatite photocatalysts: New environmentally materials to mitigate pollutants(Minerals-1634926)”. We agree with the comments and suggestions, which have been of great assistance in improving the quality of our paper and guiding our research. We have revised the paper after carefully studying the reviewer’s comments and have responded to them point by point. Revisions to the manuscript are highlighted in red. We have also made several further changes after carefully rereading the manuscript, which are likewise highlighted in red. 

We have tried our best to improve the manuscript. These changes do not influence the content and framework of the paper. We appreciate the editors’ and reviewers’ work, and hope that the revisions and accompanying responses will make our manuscript suitable for publication in this journal.  Once again, thank you very much for your comments and suggestions.

Yours sincerely, 

Dr  Josy Anteveli Osajima

Round 2

Reviewer 2 Report

General comment

The Author have addressed the majority of the raised concerns, although some key points were not addressed. Now the review is a bit less dispersive and more on focus, but still far from being completely clear. Finally, the language has not been improved significatively, and still is of difficult comprehension.

Specific comments related to points raised

  • The title is still grammatically incorrect.
  • The whole abstract is of difficult comprehension
  • Typos are very common (e.g. “hydroxyapatite”)
  • In my opinion, focusing only on research articles about hydroxyapatite is limitative. The Authors claim that there are no pertinent works/patents on photodegradation with “apatite”/“calcium phosphates”/other calcium phosphate phases, but do not show any data in merit
  • The authors has misinterpreted my comment on the “screening” sections. If they wanted to talk about how they screened the articles/patents, all the second part (“several articles….photocatalytic degradation”) is not a screening, is a description of the articles. So, it should be moved in the main part of the review
  • Since all patents are out of scope, Table 3 is unnecessary; you can only keep the explanation why there are no patents
  • Table 4 still seems out of scope. If is not related do photocatalysis, why insert it?
  • Figure 4 is confusing and does not convey a message. On the other hand figure 6 is very vague
  • Figure 5 mentions antimicrobial activity, in contrary to the caption…

Author Response

Dear Reviewer 2,

Thank you for your careful review and valuable advice! We really appreciate reviewer’s efforts towards our manuscript. We believe your suggestions will improve our manuscript to a higher level and we have again carefully checked our manuscript and made proper improvement.

We have tried our best to improve the manuscript. These changes do not influence the content and framework of the paper. We appreciate the editors’ and reviewers’ work, and hope that the revisions and accompanying responses will make our manuscript suitable for publication in this journal.  Once again, thank you very much for your comments and suggestions.

Yours sincerely, 

Dr  Josy Anteveli Osajima

Round 3

Reviewer 2 Report

The manuscript is now acceptable. There are typos and errors (e.g. a table is misplaced, out of the page, and unreadable) but I think those will be addressed by publisher team.